# Energy-Efficient Uplink Scheduling in Narrowband IoT

**DOI:** 10.3390/s22207744

**Published:** 2022-10-12

**Authors:** Farah Yassine, Melhem El Helou, Samer Lahoud, Oussama Bazzi

**Affiliations:** 1Ecole Doctorale des Sciences et Technologies, Lebanese University, Hadath 11-8281, Lebanon; 2Ecole Supérieure d’Ingénieurs de Beyrouth, Saint Joseph University of Beirut, Beirut 111, Lebanon

**Keywords:** energy efficiency, link adaptation, scheduling, NB-IoT, resource allocation

## Abstract

This paper presents a detailed study of uplink scheduling in narrowband internet of things (NB-IoT) networks. As NB-IoT devices need a long battery lifetime, we aim to maximize energy efficiency while satisfying the main requirements for NB-IoT devices. Also, as the NB-IoT scheduling problem is divided into link adaptation problem and resource allocation problem, this paper investigates the correlation between these two problems. Accordingly, we propose two scheduling schemes: the joint scheduling scheme, where the two problems are combined as one optimization problem, and the successive scheduling scheme that manages each problem separately but successively. Each scheme aims to maximize energy efficiency while achieving reliable transmission, satisfying delay requirements, and guaranteeing resource allocation specifications. Also, we investigate the impact of the selected devices to be served on the total energy efficiency. Accordingly, we propose two device selection techniques to maximize the total energy efficiency. The first technique exhaustively searches for the optimal devices, while the second sorts the devices based on a proposed priority score. The simulation results compare the successive and the joint scheduling schemes. The results show that the joint scheme outperforms the successive scheme in terms of energy efficiency and the number of served devices but with higher complexity. Also, the results highlight the impact of each proposed selection technique on the scheduling schemes’ performance.

## 1. Introduction

The third generation partnership (3GPP) has introduced the narrowband internet of things (NB-IoT) technology in its Release 13 to achieve massive and reliable connections for IoT [1,2,3]. NB-IoT is a low power wide area (LPWA) technology designed to effectively serve massive machine type communications (mMTC) requirements: low device cost, long battery life, long coverage range and a massive number of devices of high reliability [4].

NB-IoT inherits the main characteristics: framing structure, downlink and uplink transmission modulation schemes, and transmission mode from 3GPP long-term evolution (LTE) systems. NB-IoT enhances radio resource allocation. Specifically, NB-IoT supports a unique allocation specification known as resource unit (RU) that enables the allocation of specific tones or subcarriers to a device in the uplink instead of the entire resource block [5]. This allows allocating a small amount of bandwidth to a device leading to more efficient spectrum usage. In addition, NB-IoT exploits repetitions that retransmit the data to ensure reliable transmission while serving devices located at far and challenging positions [6].

With these unique specifications of NB-IoT of RU and repetitions, existing uplink scheduling schemes are not applicable for NB-IoT networks. Existing LTE link adaptation and resource allocation do not consider the unique specifications of NB-IoT of repetitions determination and the RU type configuration. The uplink scheduling in NB-IoT is performed at the subcarriers (SC) or tones level rather than the entire physical resource block as in LTE. Hence, the scheduling in NB-IoT needs further investigation while considering repetition determination and RU type configuration. Also, NB-IoT supports the communication of devices that need long battery life [7]. Hence, the scheduling in NB-IoT needs to consider energy efficiency as a critical objective. Moreover, as NB-IoT enables the communication of many devices, NB-IoT scheduling needs to optimize resource utilization with high reliability. Accordingly, this paper aims to tackle these issues by providing a detailed investigation of NB-IoT scheduling in the uplink. Our goal is to maximize the system energy efficiency while considering the basic requirements for NB-IoT devices in terms of reliable transmission, delay tolerance, and resource allocation specifications. The main contributions of the paper are briefly described as follows:We study the impact of the selected devices to utilize scarce resources on the system’s energy efficiency. Accordingly, we propose two different device selection techniques to select the optimal devices to be served to enhance the system’s energy efficiency.We aim to study the correlation between the link adaptation problem and the resource allocation problem involved in NB-IoT scheduling. The transmission parameters are selected based on the device’s network conditions in the link adaptation phase. In the resource allocation phase, resources are allocated to each device by selecting the optimal transmission parameters among the feasible ones collected in the link adaptation. Accordingly, we distinguish two different scheduling optimization problems: successive and joint scheduling problems. The successive scheduling problem manages the link adaptation problem and the resource allocation problem separately but successively. The joint scheduling problem combines the two problems as one joint optimization problem. Then we propose two heuristic schemes for each scheduling problem formulated above.The simulation results compare the performance of the proposed schemes. The results also show the proposed selection techniques’ impact on the scheduling schemes’ performance.

The paper’s contributions are described in detail in Section 2.

The remainder of the paper is structured as follows. Section 2 presents the related works and the paper contributions. Section 3 gives a general overview of NB-IoT scheduling. Section 4 describes the system model. Section 5 formulates the scheduling problem. Section 6 describes the proposed uplink scheduling schemes and the selection techniques. Section 7 interprets the simulation results. Finally, Section 8 sums up the conclusion.

## 2. Related Works and Contributions

### 2.1. Related Works

Many recent works studied different aspects of NB-IoT scheduling. The authors in [8] studied the scheduling problem in NB-IoT during the power-saving mode of the devices. Their work aimed to save more energy while enabling the communication of a high number of devices with good quality of service (QoS). In [9], the authors worked on optimizing the resource allocation in the uplink in NB-IoT networks, considering devices with different mobility features. Their optimization was based on Double Deep Q-learning Network (DDQN). The authors in [10] studied the NPDCCH offset mechanism for NB-IoT resource allocation in the uplink and downlink. Their work focused on reducing the consumed radio resources.

The link adaptation was the subject of the studies in recent papers as in [11,12,13,14,15,16,17]. In [11], the authors proposed a simple and efficient link adaptation procedure for NB-IoT. The proposed procedure determines the optimal combination of modulation and coding scheme (MCS) and the minimal number of repetitions while guaranteeing the block error rate (BLER) requirements. This ended up with a reduced number of narrowband physical uplink shared channel (NPUSCH) and narrowband physical downlink control channel (NPDCCH) transmissions. Also, the authors in [12] presented a dynamic link adaptation procedure for uplink that consists of inner and outer loop link adaptation for MCS selection and repetition determinations. The authors in [13] presented a link adaptation procedure considering a look-up table scheme that selects the optimal modulation and coding level and the number of repetitions satisfying the target quality of service (QoS) while decreasing resource consumption. The authors in [14] proposed a 2D link adaptation procedure that selects the suitable MCS level and repetition number determination saving the power and increasing the spectral efficiency for low complexity MTC. However, neither of [11,12,13] or [14] examined the RU type allocation as an additional important dimension for uplink link adaptation. The works [15,16] considered both bandwidth allocation and repetition determination in addition to the MCS selection in their link adaptation procedures. The former studied the influence of repetitions, RU type, and MCS using derived analytical expressions. It also provided an uplink link adaptation procedure that selects the optimal combination of repetitions, type of RU, and MCS that minimizes transmission time. The latter designed the hybrid link adaptation procedure for NB-IoT systems that selects the optimal values of the repetition, MCS, and bandwidth while maintaining reliability and achieving the optimal uplink latency and coverage. However, their analysis was done considering ideal channel estimation for simplicity. Our previous works in [17,18] investigated the link adaptation problem in NB-IoT. Ref. [17] focused on the impact of each link adaptation parameter, i.e., type of RU, number of repetitions, and MCS level on the transmission. In [18], we extended the investigation provided in [17]. In [18], the investigation was presented at the link level focusing on the impact of the data size and the received power on the NB-IoT performance.

The resource allocation in NB-IoT was studied in [19,20,21,22]. In [19], the authors proposed an iterative resource allocation procedure for NB-IoT using the cooperative approach considering a multi-cell cellular network. The proposed procedure aims to maximize the system rate while handling the inter-cell interference to improve the resource allocation among the users. However, the allocation was done at the level of the whole resource block, like in LTE, without considering the different tones allocation. Yu et al. investigated in [20] the uplink resource allocation in NB-IoT. They formulated the problem and then proposed a procedure that allocates to all the devices the optimal resource unit type that minimizes the number of consumed subframes. However, the selected optimal resource unit type was the same for all the devices within the same scheduling period. In [21], an NB-IoT energy efficient scheduling procedure was proposed that minimizes the consumed energy per device while enhancing the spectrum utilization. However, the authors did not consider the scheduling delay parameter between the downlink and uplink transmissions. In [22], the authors studied and formulated the resource allocation problem for NB-IoT while investigating the tradeoff between the data rate and latency. Then, they proposed a suboptimal algorithm that allocates the radio resources at the subcarriers (SC) level and the transmission power among the devices while maximizing the data rate. The procedure selects the resource unit type that best fits the remaining resources and gives a higher data rate. However, the study focused only on the resource unit type without considering the remaining scheduling parameters.

### 2.2. Contributions of Our Work

In this paper, we study the NB-IoT scheduling problem considering the radio parameters, namely RU type, MCS, repetition number, resource assignment, scheduling delay, and subcarrier set. These parameters ensure reliable transmission while satisfying delay and resource allocation constraints. The objective of the scheduling problem is to maximize the overall energy efficiency. In this section, we highlight the major contributions of this paper in comparison with state of the art.

In comparison with [19,20,21,22], this paper considers the complete scheduling parameters set of the RU type, MCS, repetition number, resource assignment, scheduling delay, and subcarrier set. The allocation done in [19] neglected the RU type allocation. In [20], the allocated RU type was the same for all the devices served in a scheduling period. In [21], the scheduling delay value was ignored. In [22], the authors focused only on the RU type allocation without highlighting the allocation of the remaining parameters specified by the BS at each scheduling instant. Table 1 shows the difference in the supported scheduling parameters of the related works.
sensors-22-07744-t001_Table 1Table 1Supported scheduling parameters.Related WorksRU TypeMCSRepetition NumberResource AssignmentScheduling DelaySubcarrier Set[19]NOYESYESYESYESYES[20]NOYESYESYESYESYES[21]YESYESYESYESNONot Mentioned[22]YESNot MentionedNot MentionedNot MentionedNot MentionedNot MentionedThis paperYESYESYESYESYESYESIn [19], the delay and resource allocation constraints were not considered. The authors in [20] and [22] did not take into consideration the delay constraints. As NB-IoT supports latency-critical applications, in this paper, we focus on the main requirements of the NB-IoT devices of reliable transmission, delay constraint, and resource allocation constraints. Table 2 shows the difference in the supported constraints of the related works.
sensors-22-07744-t002_Table 2Table 2Supported constraints.Related WorksReliable TransmissionDelay ConstraintResource Allocation Constraint[19]YESNONO[20]YESNOYES[21]YESYESNot Mentioned[22]YESNOYESThis paperYESYESYESAs the scheduling problem is composed of a link adaptation problem and a resource allocation problem, we aim to study the correlation between these two problems. Accordingly, we propose two scheduling schemes: successive and joint scheduling schemes. The successive scheduling scheme successively manages the link adaptation problem and resource allocation problem. However, the joint scheduling scheme manages the two problems as one joint problem. To our knowledge, the comparison between the joint scheme and the successive scheme is not been previously investigated and represents one of the main contributions of our paper.Also, we aim in our work to optimize the selection of the devices to be served as they affect the system energy efficiency. Accordingly, we propose two different device selection techniques. The first technique provides an exhaustive search for the optimal devices to be served such that the total energy efficiency is maximized. In the second technique, we propose a priority score that determines the order of the devices to be served to enhance the overall energy efficiency. To our knowledge, investigating the served devices’ impact on the NB-IoT scheduling’s overall performance is another significant contribution to our work as it has not been investigated before.

## 3. General Overview on NB-IoT Scheduling

In this section, we first present the signals and channels of NB-IoT, highlighting the channels involved in the scheduling problem. Then, we describe the uplink scheduling along with a simple illustrating example.

### 3.1. NB-IoT Signals and Channels

The data transmission of the NB-IoT system is carried over a bandwidth of 180 kHz. The single carrier frequency-division multiple access (SC-FDMA) is adopted in the NB-IoT uplink transmission with a subcarrier spacing of 15 kHz, including 12 subcarriers in each time slot (TS) over 1 ms. It is also possible to have subcarrier spacing of 3.75 kHz, including 48 subcarriers for TS spanning 2 ms.

Figure 1 describes a simplified NB-IoT frame structure. One frame is divided into 10 subframes, each of length 1 ms. A subframe is divided into two timeslots [2]. Multiple uplink and downlink signals and channels are periodically transmitted in NB-IoT frame structure [6].

Concerning the downlink frame, the narrowband physical broadcast channel (NPBCH), the narrowband primary synchronization signal (NPSS), and the narrowband secondary synchronization signal (NSSS) occupy subframes number 1, 6, and 10, respectively. The remaining subframes are dynamically occupied by either the narrowband physical downlink shared channel (NPDSCH) or the narrowband physical downlink control channel (NPDCCH). The NPDSCH carries the downlink data for a specific device, whereas the NPDCCH delivers the downlink control information (DCI) to the UE. Three DCI formats are distinguished for NB-IoT. DCI format N0 is involved in the NB-IoT uplink scheduling for UL Grant. As for the uplink, only two channels are supported, the narrowband physical random access channel (NPRACH) and the narrowband physical uplink shared channel (NPUSCH). After the NPRACH allocation, the NPUSCH occupies the remaining available resources. The NPRACH is used to initiate the random access (RA) procedure, whereas the NPUSCH is used for the uplink data transmission.

### 3.2. NB-IoT Uplink Scheduling

NB-IoT is designed to enable the communication of thousands of devices in different reliability-and latency-critical IoT applications: smart metering, smart city infrastructure management (e.g., traffic light management and waste management), personal e-health management, intruder and fire alarm detection. The scheduling in NB-IoT needs to manage and optimize the access of the devices on the network.

A NB-IoT device must monitor successive NPDCCH subframes called the search spaces to decode one DCI. The DCI contains the scheduling information, determined by the base station (BS), for its uplink data upload. In general, each UE can receive up to one DCI, which is carried by one NPDCCH subframe. The parameter $R_{max}$ determines the number of NPDCCH subframes which is the length of the NPDCCH search space. The time interval between two successive NPDCCH opportunities is referred to as an NPDCCH period (NP). The uplink scheduling procedure is performed every NP. The NP length is determined by $R_{max}\cdot G$ where *G* is a system parameter [3,23]. Also, the NP length determines the number of NPUSCH. The length of the NP can be adapted, which is known as NPDCCH Period Adaptation Problem addressed in [24]. In this paper, we assume that the NP length is fixed as in [20].

In order to reduce the transmission errors, the transmission of a DCI corresponding to one device is repeated $r_{DCI}$ times [3]. *Candidates* denote the NPDCCH subframes allocated for one DCI transmission. The number of candidates within an NP is denoted by ε, where $\varepsilon =R_{max}/r_{DCI}$.

The DCI carries the uplink scheduling result for a specific device, as determined by the base station (BS). The main uplink fields of the scheduling parameters and the corresponding values carried in the DCI are summarized in Table 3 [3].

#### 3.2.1. Subcarrier Indication Field ($I^{sc}$)

The RU is the smallest unit mapping an uplink transport block. NB-IoT supports multiple types of RUs: 1, 3, 6, and 12-tone with a subcarrier spacing of 15 kHz. Hence, NB-IoT supports both single-tone and multi-tone transmissions in the uplink. However, only single-tone transmission is supported for subcarrier spacing of 3.75 kHz. This paper considers the mandatory subcarrier spacing in the standard, which is 15 kHz. Thus, the BS can select one of the RU types (*y*): 1-tone, 3-tone, 6-tone, and 12-tone. The RU type is identified by the integer variable *q*:(1)$$q=\left\{\begin{array}{c} 1,\;y=12\text{-tone}\\ 2,\;y=6\text{-tone}\\ 3,\;y=3\text{-tone}\\ 4,\;y=1\text{-tone} \end{array}\right..$$

A device *i* can only be allocated one RU type for an uplink transmission within an NP *p*. The allocation of the RU type is related to the device’s radio conditions and the scheduling objective. We denote by $v_{p,i,q}\in \left\{0,1\right\}$ a binary variable that indicates if a device *i* is allocated RU type *q* such that:(2)$$\sum_{q=1}^{4}v_{p,i,q}\leq 1,\forall p\in P,\forall i\in N.$$

*N* denotes the number of devices. The RU type determines the number of subcarriers β and the duration of the RU *t* as expressed in (Equation 3) and (Equation 4) below:(3)$$\beta_{p,i}=12\cdot v_{p,i,1}+6\cdot v_{p,i,2}+3\cdot v_{p,i,3}+1\cdot v_{p,i,4}$$
(4)$$t_{p,i}=1\cdot v_{p,i,1}+2\cdot v_{p,i,2}+4\cdot v_{p,i,3}+8\cdot v_{p,i,4}.$$

Table 4 summarizes the possible RU types with their corresponding number of subcarriers and timeslots for subcarrier spacing of 15 kHz. Equations (Equation 3) and (Equation 4) guarantee the constraints on the RU type at the number of subcarriers and duration of RUs. As the value of $v_{p,i,q}$ is equal to 1 for only one *q* ensured in (Equation 2), the total number of subcarriers $\beta_{p,i}$ in (Equation 3) and the total duration $t_{p,i}$ in (Equation 4) for one RU meet the RU characteristics shown in Table 4 for each user.

The value of $I^{sc}$ indicates the allocated RU type *q* and its corresponding set of the allocated subcarriers $n^{sc}$ as shown in Table 5. For example, if $I^{sc}\in \left\{0,1,2,\ldots ,11\right\}$, i.e., the RU type is 1-tone (single-tone), there are 12 possible locations for the RU.
(5)$$n^{sc}\in \left\{\left\{0\right\},\left\{1\right\},\left\{2\right\},\left\{3\right\},\left\{4\right\},\left\{5\right\},\left\{6\right\},\left\{7\right\},\left\{9\right\},\left\{10\right\},\left\{11\right\}\right\}$$

If $I^{sc}\in \left\{12,13,14,15\right\}$, then the RU type is 3-tone with 4 possible locations.
(6)$$n^{sc}\in \left\{\left\{0,1,2\right\},\left\{3,4,5\right\},\left\{6,7,8\right\},\left\{9,10,11\right\}\right\}$$

Equations (Equation 7) and (Equation 8) below show the dependency of the subcarrier indication field value, the allocated RU type, and subcarrier sets for a device *i* in NP *p*.
(7)$$I_{p,i}^{sc}=I_{1}\cdot v_{p,i,1}+I_{2}\cdot v_{p,i,2}+I_{3}\cdot v_{p,i,3}+I_{4}\cdot v_{p,i,4}$$
(8)$$\begin{array}{c} \begin{array}{cc} n_{p,i}^{sc}= & \left\{0,1,\ldots ,11\right\}\cdot v_{p,i,1}\\ \; & +\left(6\cdot \left(I_{p,i}^{sc}-16\right)+\left\{0,1,2,3,4,5\right\}\right)\cdot v_{p,i,2}\\ \; & +\left(3\cdot \left(I_{p,i}^{sc}-12\right)+\left\{0,1,2\right\}\right)\cdot v_{p,i,3}\\ \; & +I_{p,i}^{sc}\cdot v_{p,i,4} \end{array} \end{array}$$

#### 3.2.2. Repetition Number Field ($I^{rep}$)

Repetitions in NB-IoT enable the BS to reach and serve devices in challenging radio conditions. Repeating the transmission decreases the signal-to-noise ratio threshold, thus improving the receiver sensitivity and ending up with reliable transmission. In NB-IoT, the transmitted RUs are blindly repeated. $I^{rep}$ indicates the allocated number of repetitions *r* for the uplink transmissions according to Table 6.

#### 3.2.3. Resource Assignment Field ($I^{RU}$)

$I^{RU}$ determines the number of continuously allocated RUs (*s*) as shown in Table 7 for a specific device without including the repetition. In order to obtain the total number of RUs allocated for the complete transmission, we multiply the associated resource assignments value with the allocated number of repetitions.

#### 3.2.4. Modulation and Coding Scheme Field ($I^{MCS}$)

$I^{MCS}$ determines the modulation and coding scheme level (*m*). NB-IoT supports 11 different MCS level {0∼10} for single-tone transmissions and 14 different levels {0∼13} for the multi-tone transmissions. Combining the MCS with the resource assignments determines the transport block size (TBS). MCS Level is adapted based on the radio conditions of the device. Low MCS levels are preferable for devices encountering bad radio conditions. However, for devices with good channel qualities, allocating higher MCS levels is recommended to get higher data rates.

#### 3.2.5. Scheduling Delay Field ($I^{delay}$)

$I^{delay}$ indicates, according to Table 8, the scheduling delay $k_{0}$, which is the separation time between the last DCI corresponding to the device and the start uplink subframe allocated to the device. During the scheduling delay, the device can decode the DCI message and switch from the reception mode to the transmission mode. The first NPUSCH subframe ($sb_{p,i}$) allocated to device *i* in NP *p* is determined based on the allocated scheduling value $k_{0}$ and the last NPDCCH subframe ($n_{p,i}$) of its corresponding candidate according to the following equation:(9)$$sb_{p,i}=k_{0_{i}}+n_{p,i}+1.$$

### 3.3. Scheduling Illustration

In this subsection, we present a simple example to illustrate the scheduling process in NB-IoT described in Figure 2. We suppose that $r_{DCI}=2$, the DCI transmission for each device is repeated twice. We only consider the allocation for two devices for simplicity. The candidate transmitting the DCI of device 1 occupies NPDCCH subframes 2 and 3, while NPDCCH subframes 4 and 5 carry the DCI for device 2. Accordingly, the start uplink subframe ($sb_{1}$) for device 1 is 12 (8+3+1) assuming that the allocated scheduling delay value ($k_{0_{1}}$) is 8 ms and the last NPDCCH subframe ($n_{1}$) of its candidate is subframe number 3. The number of repetitions, the resource assignment, and the RU type for device one equal 1, 1, and 6-tone, respectively. However, device 2 uploads its data on the uplink subframe ($sb_{2}$) number 14 (8+5+1) considering the scheduling delay value ($k_{0_{2}}$) 8 ms. Device 2 is allocated a RU with 12-tone combined with the number of repetitions one and resource assignment 1.

## 4. System Model

We consider an NB-IoT cellular network consisting of one BS in a circular geographical area of radius *R* and *N* devices that are uniformly distributed. We assume that 50% of the devices are located indoors. Each device denoted by *i*, where i∈{1,…,N}, sends an uplink data of size $D_{i}$ (in bits) to the BS with a transmit power $p_{i}$ where $p^{min}\leq p_{i}\leq p^{max}$. $p^{min}$ and $p^{max}$ are the minimum and the maximum transmit powers, respectively. Each device has a strict delay deadline denoted by $d_{i}$ (expressed in milliseconds). The data of the device must be uploaded before $d_{i}$.

The BS should allocate an MCS level ($m_{i}$), resource assignment ($u_{i}$), number of repetitions ($r_{i}$), scheduling delay value ($k_{0_{i}}$), RU type ($q_{i}$) and the subcarrier set ($n_{i}^{sc}$) for each device *i*. These are called the scheduling parameters and are carried out by the DCI. This problem is called the NB-IoT scheduling problem and is triggered once every NP. The NB-IoT scheduling problem has to allocate each device *i*, the optimal combination of the scheduling parameters while ensuring reliable communication transmission, satisfying delay requirements, and respecting the allocation constraint.

Reliable communication is achieved as long as the received signal to noise and interference ratio $S_{rx}$ is greater than or equal to the minimum signal to noise ratio $S_{req}$ required for successful decoding of the received uplink transmission.

The received SNR is expressed in (Equation 10). $G_{r}$, $G_{t}$, $L_{path}$, $L_{shadow}$, $L_{penetration}$, γ, *w*, *I* and $N_{0}$ denote the receiver antenna gain, the transmitter antenna gain, the path loss, the shadow fading effect, the penetration loss, indoor/outdoor binary indicator, bandwidth of the RU, interference level and the noise respectively.
(10)$$S_{rx}\left(q\right)=\frac{p\cdot G_{r}\cdot G_{t}}{L_{path}\cdot L_{shadow}\cdot \gamma L_{penetration}\cdot \left(w\cdot N_{0}+I\right)}$$

In this paper, we are estimating the $S_{req}$ while considering the cross-subframe channel estimation [25] using (Equation 11). Cross-subframe channel estimation improves the channel estimates by averaging the channel estimates over multiple consecutive subframes. Thus, the cross-subframe technique can reduce the coverage extension limitations caused by the channel estimation error σ existing in the realistic channel estimations.
(11)$$\left(2^{\frac{R_{b}}{w\cdot B_{eff}}}-1\right).S_{eff}=\frac{r.(\sigma +S_{req})}{\left(\sigma +1+\frac{\sigma }{S_{req}}\right)\left(1+\frac{\sigma }{2.S_{req}}\right)}$$

$R_{b}$ denotes the transmission bit rate measured in bits/s. *w* represents the bandwidth of the RU. $B_{eff}$ and $S_{eff}$ are the bandwidth efficiency and SNR efficiency of NB-IoT technology, respectively, that are obtained through curve fitting in the 3GPP link-level simulation results [26,27].

The estimation error σ and $S_{req}$ are linearly dependent as in (Equation 12) where $c_{1}$ and $c_{2}$ are constants obtained through link-level simulations [27].
(12)$$\sigma_{dB}=c_{1}.S_{req,dB}+c_{2}$$

$R_{b}$ is expressed as:(13)$$R_{b}\left(u,t,m\right)=\frac{B+C}{u\cdot t},$$
where *C* is the cyclic redundancy check code size (bits) and *B* is the transport block size (bits). *u* and *t* are the resource assignment and the durations of the RU, respectively.

The maximum TBS, $B_{max}$, is directly linked to the resource assignment *u* and the selected MCS *m*. $B_{max}$ is defined at the MAC layer ranging from 2 bytes (16 bits) up to 317 bytes (2536 bits) and 217 bytes (1736 bits) for multi-tone and single-tone transmissions, respectively, as specified by the 3GPP Release 14 standard [3]. The application payload and higher layer protocol overhead affect the size of *B* that must be lower than or equal to $B_{max}$ as
(14)$$B\leq B_{max}\left(u,m\right).$$

## 5. Problem Formulation

Scheduling in NB-IoT requires the BS to select for the UEs’ the optimal combination of the following parameters: the number of repetitions *r*, resource assignments *u*, MCS level *m*, scheduling delay $k_{0}$, the RU type *q*, and the subcarrier set $n_{sc}$. The selection is made to maximize the system energy efficiency while satisfying the reliable transmission, delay requirements, and allocation constraints.

### 5.1. Scheduling Objective

The energy efficiency for a device is defined as the data rate ratio over the consumed energy and is expressed in (Equation 15). DR denotes the data rate, which is the data size ratio over the transmission duration as in (Equation 16). CE denotes the consumed energy, which is the transmitted power multiplied by the transmission duration as in (Equation 17).
(15)$$EE\left(r,u,m,q\right)=\frac{DR(r,u,m,q)}{CE(r,u,m,q)}=\frac{D}{p\cdot {\left(u\cdot t\cdot r\right)}^{2}}$$
(16)$$DR\left(r,u,m,q\right)=\frac{D}{u\cdot t\cdot r}$$
(17)CE(r,u,m,q)=p·u·t·r

The transmission duration is the number of repetition *r* multiplied by the duration of the RU *t* and the resource assignment *u*.

### 5.2. Scheduling Constraints

To guarantee feasible uplink scheduling, the following constraints must be satisfied.

#### 5.2.1. Reliable Transmission Constraint

Communication is reliable if the received signal-to-noise and interference ratio $S_{rx}$ is greater than or equal to the minimum signal-to-noise ratio $S_{req}$:(18)$$S_{rx}\geq S_{req}.$$

#### 5.2.2. Delay Constraint

The delay deadline of a device *i* must be satisfied where the data transmission to the BS is accomplished before $d_{i}$. The delay constraint is expressed in the following:(19)$$sb\left(k_{0}\right)+u\cdot r\cdot t\leq d_{i}.$$

In the previous equation, u·r·t is the uplink transmission duration. In this paper, we assume that the sb must be within the current NP of length *L* to be considered feasible as expressed in (Equation 20).
(20)$$1\leq sb(k_{0})\leq L$$

Also, the device must be able to finish uploading its uplink data without crossing the next NP using the allocated scheduling parameters described in (Equation 21).
(21)sb+u·r·t−1≤L

#### 5.2.3. Resource Allocation Constraint

We define a binary allocation variable $x_{p,s,c,i}\in \left\{0,1\right\}$ that indicates if a device *i* is allocated subcarrier *c* at subframe *s* in NP *p* or not. This resource element can be allocated at maximum for only one device to avoid overlapping devices:(22)$$\sum_{i=1}^{N}x_{p,s,c,i}\leq 1,\forall p,\forall s,\forall c.$$

The BS has to assign the device *i* in NP *p*, a number of u·r·t NPUSCH subframes as described in (Equation 23).
(23)$$\sum_{s=1}^{L}x_{p,s,c,i}=u_{i}\cdot r_{i}\cdot t_{i},\forall p,\forall i,\forall c$$

The allocated NPUSCH subframes for device *i* must be consecutive as ensured in (Equation 24).
(24)$$\sum_{s=1}^{L}\left|\left(x_{p,s,c,i}-x_{p,s+1,c,i}\right)\right|\leq 2,\forall p,\forall i,\forall c$$

Each of the allocated subframes must occupy a total number of subcarriers denoted by β of the allocated $n_{p,i}^{sc}$ as described in (Equation 25) where $c_{1}=min\;n_{p,i}^{sc}$ and $c_{end}=max\;n_{p,i}^{sc}$.
(25)$$\sum_{s=1}^{L}\sum_{c_{1}}^{c_{end}}x_{p,s,c,i}=\beta_{i}\cdot \left(u_{i}\cdot r_{i}\cdot t_{i}\right),\forall p,\forall i$$

The NB-IoT scheduling problem is divided into link adaptation and resource allocation problems. These two problems can be managed separately but successively or combined as one joint problem. Accordingly, we introduce the successive scheduling problem and the joint scheduling problem.

##### Successive Scheduling Problem (SSP)

The SSP is divided into link adaptation and resource allocation problems. In the link adaptation problem, the BS needs to select for each device the optimal combination of the resource assignment $u_{i}$, MCS level $m_{i}$, RU type $q_{i}$, and the number of repetitions $r_{i}$ that maximizes the energy efficiency while ensuring reliable transmission and satisfying the delay constraint. The general link adaptation optimization problem applied for each device *i* is formulated hereafter:(26)$$\begin{array}{c} \underset{r,u,m,q}{max}\;EE\left(r,u,m,q\right)\\ s.t\;S_{rx}\left(q\right)\geq S_{req}\left(m,u,r\right)\\ p^{min}\leq p\leq p^{max}\;\\ r\cdot t\cdot u\leq d\;\;\;\\ \;B\leq B_{max}\left(u,m\right)\; \end{array}.$$

In the resource allocation problem, the BS chooses, for the optimal combination selected in the link adaptation problem, the first feasible subcarrier set $n_{i}^{sc}$ and the scheduling delay value $k_{0_{i}}$ satisfying the allocation constraints and the delay requirements.

##### Joint Scheduling Problem (JSP)

The JSP combines the link adaptation and resource allocation problems into one joint optimization problem. In the JSP, the BS selects the optimal combination of the resource assignment $u_{i}$, MCS $m_{i}$, number of repetition $r_{i}$, RU type $q_{i}$, subcarrier set $n_{i}^{sc}$ and scheduling delay $k_{o_{i}}$ maximizing the global energy efficiency in the network while ensuring reliable transmission, and satisfying the delay and allocation constraints. The JSP is expressed as:(27)$$\underset{r,u,m,q,k_{0},n^{sc}}{max}\;EE\left(r,u,m,q\right)$$
subject to (Equation 18) to (Equation 25). The defined notations are listed in Table 9.

## 6. Uplink Scheduling Schemes and Selection Techniques

In this section, we propose two uplink heuristic scheduling schemes for NB-IoT cellular networks: the successive and joint scheduling schemes corresponding to the SSP and the JSP, respectively. The proposed schemes compute the optimal combination of the scheduling parameters to maximize the system total energy efficiency of the network while satisfying the constraints (Equation 18) to (Equation 25). Then we propose two different device selection techniques: the exhaustive search technique (EST) and the sorting score technique (SST), to optimize the selection of the served devices in each NPDCCH Period. The different proposed schemes and techniques are shown in Table 10.

### 6.1. Heuristic Scheduling Scheme

In Section 5, we introduced two scheduling problems: SSP and JSP. Accordingly, in this subsection, we distinguish two uplink heuristic scheduling schemes: the successive scheduling scheme (SSS) and the joint scheduling scheme (JSS) corresponding to the SSP and the JSP, respectively. SSS and JSS are applied at the level of each device to determine its optimal combination of the scheduling parameters. The optimal combination maximizes the system’s total energy efficiency of the network while satisfying the constraints (Equation 18) to (Equation 25). Algorithms 1 and 2 show the pseudocode for the SSS, and the JSS, respectively, applied for each device.

Algorithm 1 shows the pseudocode of the SSS. Step 1 executes the Algorithm 3 to get all the feasible combinations of the link adaptation parameters for a device. The obtained combinations are stored in set L. Step 2 selects the optimal link adaptation combination denoted by $l^{*}$ among all the stored ones in L. $l^{*}$ is the feasible link adaptation combination maximizing the energy efficiency. In step 3, Algorithm 4 is executed to obtain all the feasible scheduling delay values $k_{0}$ and subcarrier sets $n^{sc}$ combined with the optimal selected combination $l^{*}$. Finally, in steps 4, 5 and 6 the first feasible $k_{0}$ and $n^{sc}$ are selected combined with the optimal link adaptation combination $l^{*}$ to form $h^{*}$.
**Algorithm 1:** SSS
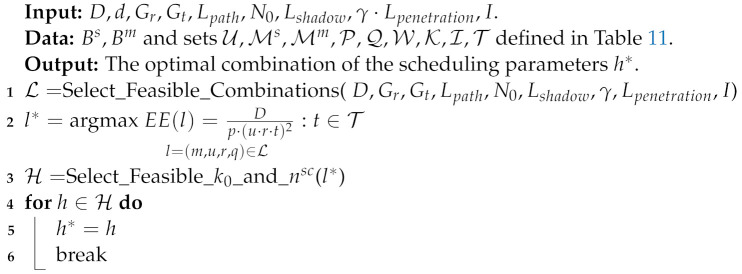

sensors-22-07744-t011_Table 11Table 11Data set Descriptions.Data SetDescriptionU={1,…,10}Resource AssignmentK={8,16,32,64}Scheduling delay values in msI={0,…,18}subcarrier indication valuesMs={0,…,10}MCS for single-toneMm={0,…,13}MCS for multi-toneP={1,2,4,8,16,32,64,128}Number of repetitionsW={180,90,45,15}RU bandwidth in kHzQ={1,2,3,4}RU typeT={1,2,4,8}RU duration in msBsMaximum TBS for single-toneBmMaximum TBS for multi-tone

Algorithm 2 presents the pseudo-code of the JSS. Step 1 obtains all the feasible combinations of the link adaptation parameters for a device by executing the Algorithm 3. The obtained combinations are stored in set L. Step 2 executes the Algorithm 4 to obtain all the feasible scheduling delay values $k_{0}$ and subcarrier sets $n^{sc}$ combined with each link adaptation combination in L. All the feasible scheduling combinations are stored in set H. Finally, step 3 selects the optimal scheduling combination denoted $h^{*}$ among all the ones stored in H. $h^{*}$ is the link adaptation combination maximizing the energy efficiency while combined with the first feasible scheduling delay value $k_{0}$ and subcarrier set $n^{sc}$. Note that $k_{0}$ and $n^{sc}$ have no impact on energy efficiency. The resource allocation phase selects the feasible link adaptation combinations at the level of the resources. Suppose at least one pair of $k_{0}$ and $n^{sc}$ satisfies the constraints when combined with a link adaptation combination. In that case, this link adaptation combination is considered feasible at the level of the resources.
**Algorithm 2:** JSS  **Input:**
$D,d,G_{r},G_{t},L_{path},N_{0},L_{shadow},\gamma ,L_{penetration},I$.  **Data:**
$B^{s},B^{m}$ and sets $U,M^{s},M^{m},P,Q,W,K,I,T$ defined in Table 11.  **Output:** The optimal combination of the scheduling parameters $h^{*}$.**_1_**L=Select_Feasible_Combinations($D,G_{r},G_{t},L_{path},N_{0},L_{shadow},\gamma ,L_{penetration},I$)**_2_**H=Select_Feasible_$k_{0}$_and_$n^{sc}$(L)**_3_**$h^{*}=\underset{h=(m,u,r,q,k_{0},n^{sc})\in H}{\mathrm{argmax}EE\left(h\right)=\frac{D}{p\cdot {\left(u\cdot r\cdot t\right)}^{2}}:t\in T}$

The Algorithm “Select_Feasible_Combinations” gets all the feasible combinations of the link adaptation parameters of MCS *m*, the number of repetitions *r*, resource assignments *u*, and the RU type *q* for a device. The feasible pairs of MCS level *m* and the number of RUs *u*, whose corresponding maximum TBS *B* is greater than the user’s data size *D*, are first collected and stored in sets $F^{s}$ and $F^{m}$ respectively for single-tone and multi-tone transmissions as expressed in lines 1 and 2. Then, the required SINR for single-tone $S_{req}^{s}$ and multi-tone $S_{req}^{m}$ are estimated respectively for each pair ∈$F^{s}$ and ∈$F^{m}$ combined with each number of repetition *r* as in lines 3 to 6.

In steps 7, 8, and 9, the received SINR $S_{rx}$ is computed assuming that the device is transmitting at the maximum power $p^{max}$ for different types of RU. All the feasible combinations of *m*, *u*, *r* and *q* ensuring reliable transmission and satisfying delay constraint in addition to their corresponding minimum-feasible transmit power *p* are collected in set L as in line 10. Note that *p* is obtained considering the received $S_{rx}$ equal to the required $S_{req}$.


**Algorithm 3:**


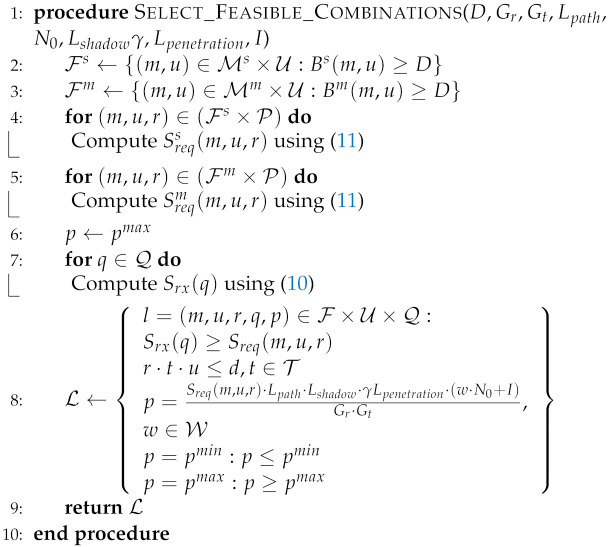



The Algorithm “Select_Feasible$\_k_{0}\_$and$\_n^{sc}$” collects all the feasible scheduling delay values $k_{0}$ and subcarrier sets $n^{sc}$ for each link adaptation combination stored in set X. Each combination in X is combined with each feasible scheduling delay $k_{0}$ and subcarrier set $n^{sc}$ guaranteeing the resource allocation constraints and the delay constraints, then stored in set H.


**Algorithm 4:**

  **procedure** SELECT_FEASIBLE_*k*_0__AND_*n^sc^*(X)
2:   $H\leftarrow \left\{\begin{array}{c} h=\left(m^{*},u^{*},r^{*},q^{*},k_{0},n^{sc}\right)\in X\times K\times I:\\ \sum_{i=1}^{N}x_{p,s,c,i}\leq 1\\ \sum_{s=1}^{L}x_{p,s,c,i}=u^{*}\cdot r^{*}\cdot t^{*},t\in T\\ \sum_{s=1}^{L}\left|\left(x_{p,s,c,i}-x_{p,s+1,c,i}\right)\right|\leq 2\\ \sum_{s=1}^{L}\sum_{c_{1}}^{c_{end\;}}x_{p,s,c,i}=\beta^{*}\cdot \left(u^{*}\cdot r^{*}\cdot t^{*}\right)\\ sb_{k_{0}}+u^{*}\cdot r^{*}\cdot t^{*}\leq d\\ 1\leq sb_{k_{0}}\leq L\\ sb_{k_{0}}+u^{*}\cdot r^{*}\cdot t^{*}-1\leq L \end{array}\right\}$
   **return**
H
4: **end procedure**


#### Algorithms Complexity

The general time complexity for each of SSS and JSS is O(i,c,k,s)=I·C·K·S, where *i*, *c*, *k*, and *s* denote the number of devices, the number of feasible link adaptation combinations, the number of possible scheduling delay values, and the number of possible subcarrier-sets respectively. The number of possible scheduling delay values can reach up to 4 possible values. The number of possible subcarrier sets depends on the RU type. Its value can reach 12 when a single-tone (1-tone) is selected.

The basic difference between the SSS and the JSS is in the number of feasible link adaptation combinations *C* considered from the Algorithm "Select_Feasible_Combinations". In the SSS, C=1 since only one link adaptation combination is considered, which is the one maximizing the energy efficiency. However, the selected feasible number of combinations can reach up to C=u·m·r·q=8·14·8·4=3580 in the JSS for each device. *u*, *m*, *r*, and *q* denote the possible values of the resource assignment, MCS level, number of repetitions, and the RU types, respectively. Hence, the JSS yields higher time complexity compared to the SSS.

### 6.2. Device Selection Techniques

As the number of served devices within an NP could not exceed the number of candidates ε, selecting the devices to be served in each NPDCCH Period can affect the system energy efficiency. Accordingly, in this section, we aim to optimize the selection of the devices by providing two different selection techniques: exhaustive search technique (EST) and sorting score technique (SST). The EST exhaustively searches for the optimal devices to be served, considering all possible combinations of the devices with different permutations. The selected optimal served devices achieve the highest total energy efficiency. The SST utilizes a proposed priority score to select the optimal devices to be served to enhance energy efficiency. After selecting the optimal devices considering EST or SST, the optimal combination of the scheduling parameters is obtained for each device considering the proposed heuristic scheduling schemes: SSS or JSS. First, we describe the exhaustive search technique (EST) and then the sorting score technique (SST).

#### 6.2.1. Exhaustive Search Technique (EST)

The EST exhaustively searches for the optimal devices to be served that maximize the total EE by considering the different possible combinations of the selected devices with different permutations.

Figure 3 describes the steps applied in the EST. First, EST collects all the different combinations of the devices to be served with different permutations and stores them in set *E*. Then, EST executes one of the proposed heuristic methods (JSS or SSS) to obtain the optimal combination of scheduling parameters ($h^{*}$) for each combination E∈E. Then, this technique computes the energy efficiency ($E(h^{*})$) corresponding to $h^{*}$. The total energy efficiency *T*, obtained from E, is updated. Next, the value of *M* is updated if *T* is greater than *M*. *M* stores the highest value of the obtained total energy efficiency. The optimal combination of devices denoted by $E_{opt}$ is set to the combination giving the highest *T*.

#### 6.2.2. Sorting Score Technique (SST)

Considering all possible combinations of devices makes the EST complex to implement. The SST is proposed to reduce this complexity. In the SST, we assign for each device *i* in the network a priority score $score_{p_{i}}$ described in (Equation 28). $score_{p_{i}}$ depends on the received power $p_{r_{i}}$, delay constraint $d_{i}$, the payload size $D_{i}$ and the highest energy efficiency EE that the device *i* can achieve. $score_{p_{i}}$ is expressed as:(28)$$score_{p_{i}}=a_{1}\cdot P_{d_{i}}+a_{2}\cdot P_{p_{r_{i}}}-a_{3}\cdot P_{D_{i}}-a_{4}\cdot P_{EE_{i}}$$
where $a_{i}\in \left[0,1\right],i=1,\ldots ,4$ are the weight factors such that $a_{1}+a_{2}+a_{3}+a_{4}=1$. $P_{d}$ denotes the delay priority, which is the assigned priority based on the delay constraint $d_{i}$ of device *i*. Devices with lower delay values need to be served first. We can express $P_{d_{i}}$ for device *i* as
(29)$$P_{d_{i}}=\frac{\underset{z}{max}\left(d_{z}\right)-d_{i}}{\underset{z}{max}\left(d_{z}\right)-\underset{z}{min}\left(d_{z}\right)}$$
where *z* is the index of the remaining devices.

$P_{p_{r}}$ and $P_{D}$ denote the received power priority and the data size priority, respectively. The received power priority is the assigned priority based on the received power $p_{r}$ of the device, while the data size priority depends on the data size *D* of a device. Lower received power values indicate bad radio conditions that limit the number of feasible combinations of the scheduling parameters. Lower received power values would decrease EE. Also, a larger data size would lower the number of feasible combinations. Hence, devices with lower received power or/and higher data sizes need higher priority values. We can write $P_{p_{r_{i}}}$ and $P_{D_{i}}$ as in (Equation 30) and (Equation 31).
(30)$$P_{p_{r_{i}}}=\frac{\underset{z}{max}\left(p_{r_{z}}\right)-p_{r_{i}}}{\underset{z}{max}\left(p_{r_{z}}\right)-\underset{z}{min}\left(p_{r_{z}}\right)}$$
(31)$$P_{D_{i}}=\frac{\underset{z}{max}\left(D_{z}\right)-D_{i}}{\underset{z}{max}\left(D_{z}\right)-\underset{z}{min}\left(D_{z}\right)}$$

$P_{EE}$ denotes the energy efficiency priority. Devices giving higher energy efficiencies are assigned higher energy efficiency priority. As we aim to maximize the system energy efficiency, serving devices with high energy efficiencies increase the system energy efficiency. $P_{EE_{i}}$ is expressed as
(32)$$P_{EE_{i}}=\frac{\underset{z}{max}\left(EE_{z}\right)-EE_{i}}{\underset{z}{max}\left(EE_{z}\right)-\underset{z}{min}\left(EE_{z}\right)}$$

The devices are then sorted in descending order based on the priority score $score_{p}$ and stored in set Z. Then starting from the first device in the set, the optimal combination of the scheduling parameters is obtained for each device considering each of the proposed heuristic scheduling schemes: SSS or JSS as described in Figure 4.

## 7. Performance Evaluation

In this section, we compare the performance of the successive scheduling scheme (SSS) to the joint scheduling scheme (JSS) while selecting the optimal combination of the scheduling parameters. Also, we study the impact of the proposed device selection techniques: EST and SST, on the behavior of each of the proposed scheduling schemes: SSS and JSS. More precisely, we also aim to study the impact of the proposed priority score $score_{p}$ on the performance of each of the SSS and JSS. Accordingly, we distinguish three techniques considering the SST: SST-EE, SST-SC, and SST-RD. SST-EE technique sorts the devices based on their energy efficiencies considering only $P_{EE}$ where $a_{1}=a_{2}=a_{3}=0$ and $a_{4}=1$. SST-SC sorts the devices based on the entire $score_{p}$ where $a_{1}=a_{2}=a_{3}=a_{4}=1/4$. SST-RD neglects the proposed score $score_{p}$ where $a_{1}=a_{2}=a_{3}=a_{4}=0$ and the devices are randomly sorted. Figure 5 illustrates these different device selection techniques.

We assume the devices are uniformly distributed in a circular area of radius 4 km. We suppose the packets sent from each device are stored in queues where each queue corresponds to one device. The packet arrival follows Poisson distribution considering that the packet size is 50 bytes. At each NP, the BS checks the number of packets arriving at each queue for each device, and the payload size is considered the number of packets stored multiplied by the packet size. The delay requirement is randomly assigned between 100 ms and 300 ms for each device. We used MATLAB to obtain the simulation results. Table 12 shows the simulation parameters.

To get a more accurate analysis with 95% confidence interval, we consider 50 different realizations with different radio conditions. In addition, we show the *p*-value (*p*) in our results, highlighting the statistical significance of our comparison. As a matter of fact, the lower the *p*-value, the greater the statistical significance of the observed difference. Note that the path loss is computed based on the Okumura-Hata propagation model.

### 7.1. Comparison between EST and SST

This subsection aims to compare the performance of SST and EST in terms of the average number of served devices and the distribution of the obtained energy efficiency considering each SSS and JSS. We consider a small-sized network containing 20 devices. The devices are simultaneously active and have data to transmit.

As NB-IoT aims to achieve massive connections, studying the variation in the number of served devices is considered an important performance indicator. Accordingly, Figure 6 shows the variation in the average number of served devices per realization obtained in the EST and the SST, considering each of the SSS and the JSS. The average number of served devices $t_{avg}$ is the summation of the number of served devices in each realization *r* denoted by $t_{r}$ divided by the total number of realizations *R* as shown in (33).
(33)$$t_{avg}=\frac{\sum_{r=1}^{R}t_{r}}{R}$$

We observe that the average number of served devices obtained considering the JSS (4.2, 3.375, 3.905, 3.441) is greater than that obtained in the SSS (4.001, 1.628, 2.401, 2.003) in each of the EST, the SST-EE, the SST-SC, and the SST-RD respectively. Also, we notice that the SST-SC achieves a higher number of served devices than the SST-RD and the SST-EE. However, the EST outperforms the SSTs. The EST can serve a higher number of devices as it considers the different selections of the devices and then selects the one maximizing the sum of the energy efficiency. The SST-SC gives the devices with limited feasible combinations higher priority. Most of these devices are encountering bad radio conditions and lower energy efficiencies. Thus, these devices occupy fewer frequency resources as they are more allocated to the single-tone transmission for reliable transmission. This leads to more free resources at the level of the frequency dimension for other devices. The SST-EE serves the devices with high energy efficiencies. These devices are thus allocated more frequency resources decreasing the remaining free frequency resources for other devices. Note that the maximum number of served devices per NP is 8 in our case (ε=8).

Figure 7 shows the distribution of the energy efficiency obtained from the served devices in each EST and SST considering the JSS. We observe that the energy efficiency obtained in the EST is greater than that obtained in the SST. More than 20% of the devices have energy efficiency above 0.355 Mbits/Watt·s${}^{2}$ in the EST. However, the SST-SC can serve the devices with low energy efficiency served in the EST. Around 19% of the devices obtain energy efficiency greater than 0.355 Mbits/Watt·s${}^{2}$ in SST-SC, while only around 15% of the devices have energy efficiency above 0.355 Mbits/Watt·s${}^{2}$ in SST-EE. This percentage is further decreased in the SST-RD, where around 12% of the devices achieve energy efficiency of more than 0.355 Mbits/Watt·s${}^{2}$.

Figure 8 presents the distribution of the energy efficiency obtained from the served devices in each EST and SSTs considering the SSS. We can observe that the EST outperforms the SSTs, where around 15% of the devices have energy efficiency greater than 2.5124 Mbits/Watt·s${}^{2}$ in the EST. However, around 10% and 6% of the devices have energy efficiency greater than 2.5124 Mbits/Watt·s${}^{2}$ in the SST-SC and SST-EE, respectively. We can notice that the SST-SC performs better than the SST-EE, especially at the level of the served devices with low energy efficiency.

Figure 9 reflects the variation in the time complexity of the SSS and JSS considering the SST-SC selection technique. Figure 9 represents the obtained execution times of the two schemes from the different realizations using a boxplot where the JSS achieves a higher median value (10.2 s) compared to the median achieved in the SSS (4.78 s). This difference in the achieved execution times is due to the variation in the time complexity where the time complexity of JSS is higher than that of SSS.

Based on the analysis of Figure 6, Figure 7, Figure 8 and Figure 9, we can notice that the JSS outperforms the SSS considering the different device selection techniques. The JSS can achieve higher energy efficiency and increase the number of served devices. However, the JSS achieves higher time complexity as it considers all the feasible link adaptation combinations from the “Select_Feasible_Combinations” procedure. Also, we can observe that the EST performs better than the SSTs in terms of the number of served devices and energy efficiency. Nevertheless, the EST is not applicable for real case scenarios with a large number of devices in the network due to its computational complexity. The SST is more scalable and practical. Also, at the level of the SST, we can infer that the SST-SC can improve the system energy efficiency more than the SST-EE as it considers the devices with low energy efficiency served in the EST.

### 7.2. Performance Evaluation of the SST in Different Networks

This subsection aims to study the impact of the number of active devices in the network on the performance of each SST-SC, SST-EE, and SST-RD, considering SSS and JSS. We consider three different network sizes: network size of 20 devices, network size of 100, and network size of 1000.

Figure 10 and Figure 11 show the variation in the average number of served devices and the average energy efficiency per device, respectively, in the SST-SC, SST-EE, and SST-RD considering the JSS. We can notice in Figure 10 that the average number of served devices is the highest in the SST-SC in networks of 20 devices and 100 devices. The average numbers of served devices are 6.62, 7.46, and 6.96, respectively, in the SST-EE, SST-SC, and SST-RD in network of 100 devices. However, in the network of 1000 devices, the number of served devices is maximum in all the techniques: SST-SC, SST-EE, and SST-RD, due to the high user diversity in the large size network. The maximum number of served devices within an NP is eight devices in our case (ε=8). Figure 11 shows the obtained average energy efficiency per device per realization in the different techniques. The average energy efficiency per device per realization $EE^{avg}$ is obtained by adding the average energy efficiency per device $avg_{r}^{EE}$ obtained in each realization *r*. Then, the sum is divided by the total number of realizations *R* as in (Equation 34).
(34)$$EE^{avg}=\frac{\sum_{r=1}^{R}avg_{r}^{EE}}{R}$$

The average energy efficiency per device $avg_{r}^{EE}$ for realization *r* is the sum of the obtained energy efficiency for each served device *j* in realization *r* divided by the number of served devices $t_{r}$ in the realization *r* as in (Equation 35).
(35)$$avg_{r}^{EE}=\frac{\sum_{j=1}^{t_{r}}EE_{j,r}}{t_{r}},\forall r$$

Figure 11 shows that the obtained average energy efficiency per device is the highest in the SST-EE in the different networks. The obtained average energy efficiency in the SST-EE, SST-SC, and SST-RD are 4.777·10${}^{4}$ Mbits/Watt·s${}^{2}$, 168.3 Mbits/Watt·s${}^{2}$, and 3474 Mbits/Watt·s${}^{2}$ respectively in networks of 100 devices. We can notice that the obtained average energy efficiency per device in the SST-SC is the lowest in the three networks compared to the SST-EE and SST-RD. That is because the SST-SC serves devices encountering bad radio conditions and ending up with low energy efficiency. However, such devices are allocated fewer resources at the level of the spectral dimension leading to more free frequency resources for other devices. This explains the higher number of served devices achieved in the SST-SC in the small-sized network. SST-EE serves devices with the highest energy efficiency ending up with a very high average energy efficiency per device in Figure 11. However, such devices are allocated many spectral resources ending up with few empty resources for the remaining devices. This leads to fewer served devices, as shown in Figure 10 in the small-sized network. In larger networks, we have multi-user diversity. The possibility of having devices encountering better radio conditions is higher. This explains the higher average energy efficiency per device achieved in the SST-EE network of size 1000 devices. Also, the possibility of having devices fitting in the remaining free resources is higher. This explains the higher number of served devices in the different techniques in the network containing 1000 devices. Also, we can notice that the performance of the SST-RD is similar in the three networks as expected. The SST-RD selects the devices to be served randomly without giving higher priority to devices with high energy efficiency as in SST-EE or devices with limited feasible combinations as in the SST-SC. SST-RD achieves higher average energy efficiency than the SST-SC but a lower number of served devices in the small-sized network. The SST-EE outperforms the SST-SC and the SST-RD in large-sized networks as it maximizes the total energy efficiency while serving the maximum number of served devices.

Figure 12 and Figure 13 present the variation in the average number of served devices and the average energy efficiency per device per realization, respectively, in the SST-SC, SST-EE, and SST-RD considering the SSS. Figure 12 shows that the SST-SC gives, as in JSS, the highest number of served devices in the different networks. The average number of served devices obtained in the SST-EE, SST-SC, and SST-RD are 1.16, 4.68, and 4.42, respectively, considering the network of 100 devices. Figure 13 shows that the highest average energy efficiency per device is obtained in the SST-EE in all the networks. The average energy efficiency per device is 5.082·10${}^{4}$ Mbits/Watt·s${}^{2}$, 406.3 Mbits/Watt·s${}^{2}$, and 4821 Mbits/Watt·s${}^{2}$ respectively in the SST-EE, SST-SC, and the SST-RD in the network of 100 devices. We can infer that the SST-SC performance is better than that of the SST-EE in the different networks, considering the SSS in terms of the number of served devices. The SST-EE, considering the SSS selects the combination with the highest EE for each device. The SST-EE, considering the SSS is greedy regarding spectral resources as in the JSS. In large networks, the possibility of serving device(s) with very high energy efficiency is higher in the SST-EE due to the multi-user diversity. However, due to the limited combinations selections in the SSS, as it selects the combination with the highest EE, the possibility of having a device that can fit in the remaining resources is very low. This leads to a minimal number of served devices in the SST-EE. The average numbers of served devices obtained in the SST-RD in the different networks are very close to the ones obtained in the SST-SC. However, SST-RD obtains higher average energy efficiency per device than SST-SC.

Based on the analysis of Figure 10, Figure 11, Figure 12 and Figure 13, we can infer that the SST-EE has the best performance in terms of both the number of served devices and the total energy efficiency in large networks while considering the JSS. However, in small-sized networks, SST-SC outperforms SST-EE in terms of the number of served devices while considering the JSS. The small-sized network has a smaller number of devices; thus, the probability of having devices needing a small amount of free remaining resources is low after the devices with the highest EE are served in the SST-EE. Considering the SSS, SST-SC has the best performance regarding the number of served devices in the different sized networks. That is because it considers devices with limited feasible combinations. The allocated combination for such devices occupies a small number of spectral resources ending up with a higher number of served devices. The SST-EE serves devices with very high EE. The allocated combination for such devices requires a massive amount of frequency resources ending up with a very low number of served devices. However, the SST-RD can give higher energy efficiency per device than the SST-SC while serving a similar number of devices.

## 8. Conclusions

This paper studied the NB-IoT uplink scheduling that maximizes the system’s energy efficiency while satisfying reliability, delay, and resource allocation constraints. The paper divides the scheduling problem into link adaptation and resource allocation problems and proposes two different scheduling schemes: the successive scheduling scheme (SSS) and the joint scheduling scheme (JSS). To optimize the selected devices to be served, we distinguish two device selection techniques: exhaustive search technique (EST) and sorting score technique (SST). EST exhaustively searches for the optimal devices, while the SST sorts the devices based on a proposed priority score. Simulation results show that the joint scheme is more efficient at the total energy efficiency and massive connection level than the successive scheme. However, the combinational complexity of the joint scheme is much higher than that of the successive one. Also, the EST achieves higher total energy efficiencies than the SST, yet the EST is not applicable for real-life scenarios due to the high computational complexity. Moreover, simulation results show the improvement in the performance of the scheduling schemes when considering the SST at the level of the number of served devices and the achieved energy efficiency.

## Figures and Tables

**Figure 1 sensors-22-07744-f001:**
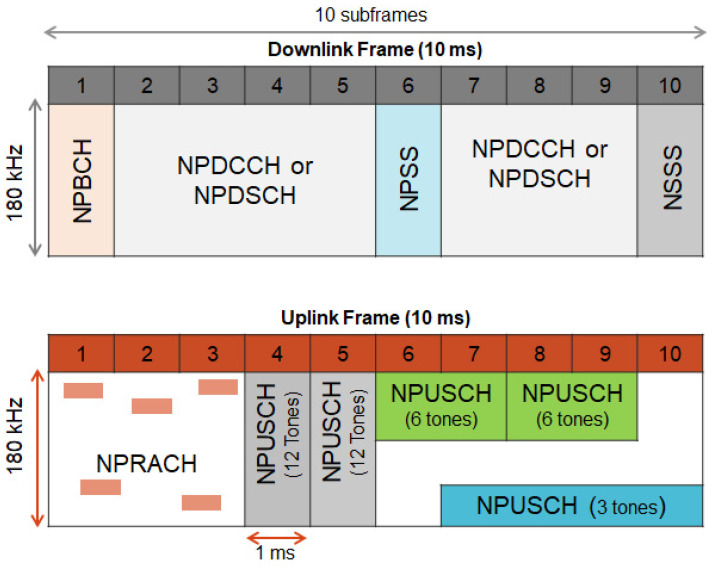
A representation of the NB-IoT uplink and downlink frames structure considering 15 kHz subcarrier spacing in uplink.

**Figure 2 sensors-22-07744-f002:**
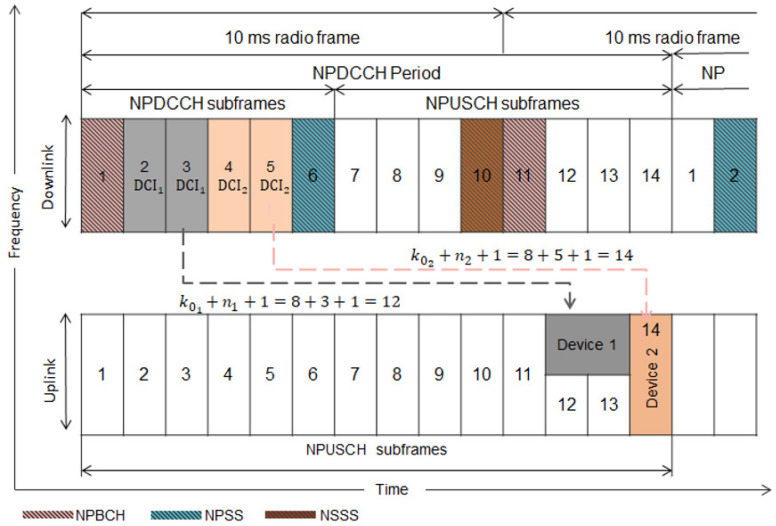
Scheduling Illustration Example.

**Figure 3 sensors-22-07744-f003:**
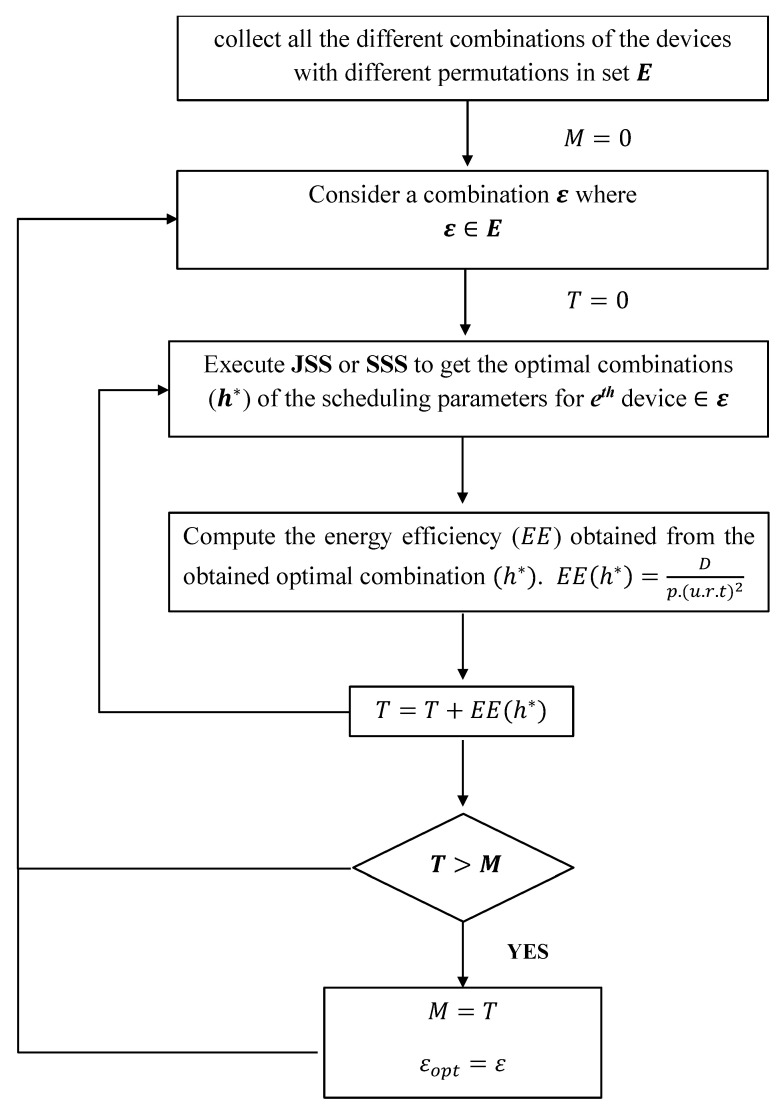
Block diagram describing steps of the exhaustive search technique (EST).

**Figure 4 sensors-22-07744-f004:**
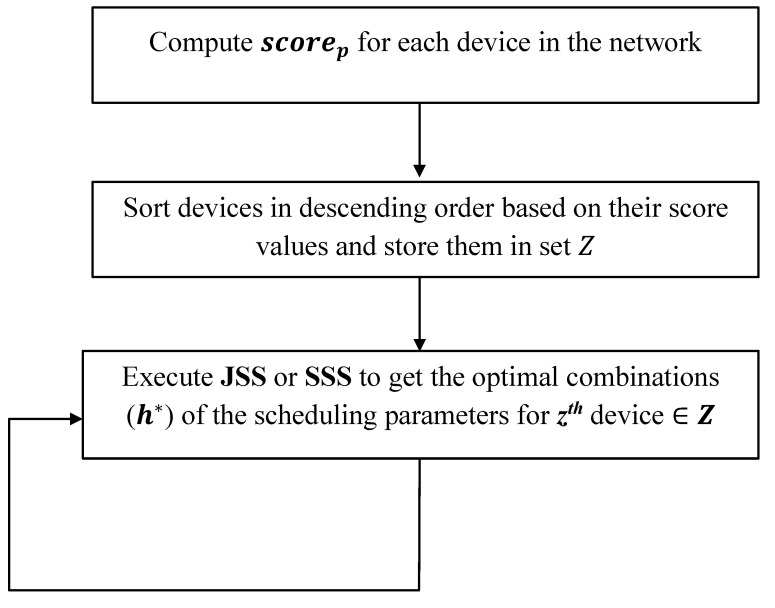
Block diagram describing steps of the sorting score technique (SST).

**Figure 5 sensors-22-07744-f005:**
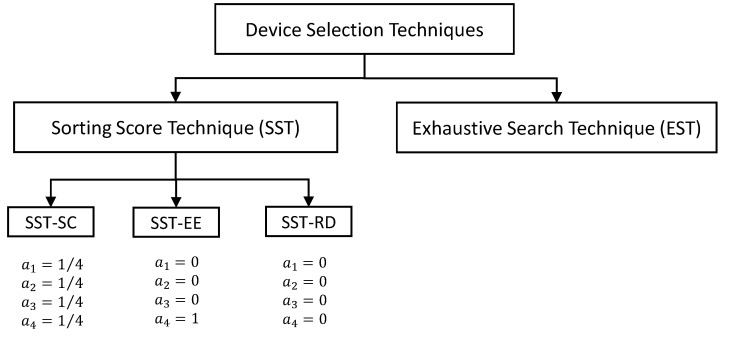
Block diagram describing the different device selection techniques.

**Figure 6 sensors-22-07744-f006:**
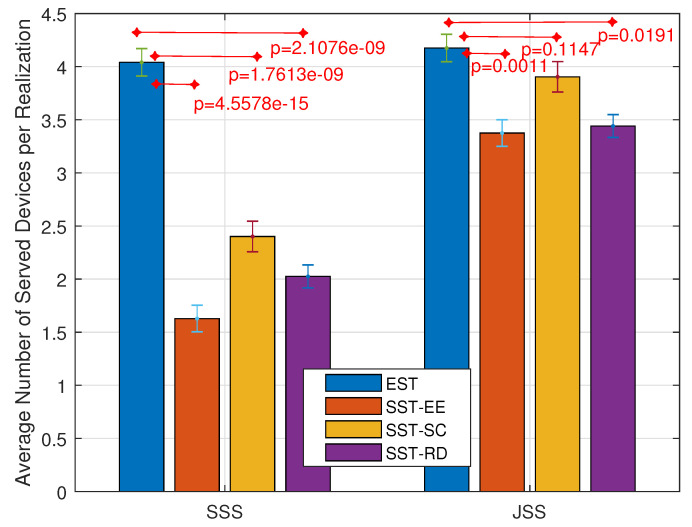
Variation in the average number of served devices per realization.

**Figure 7 sensors-22-07744-f007:**
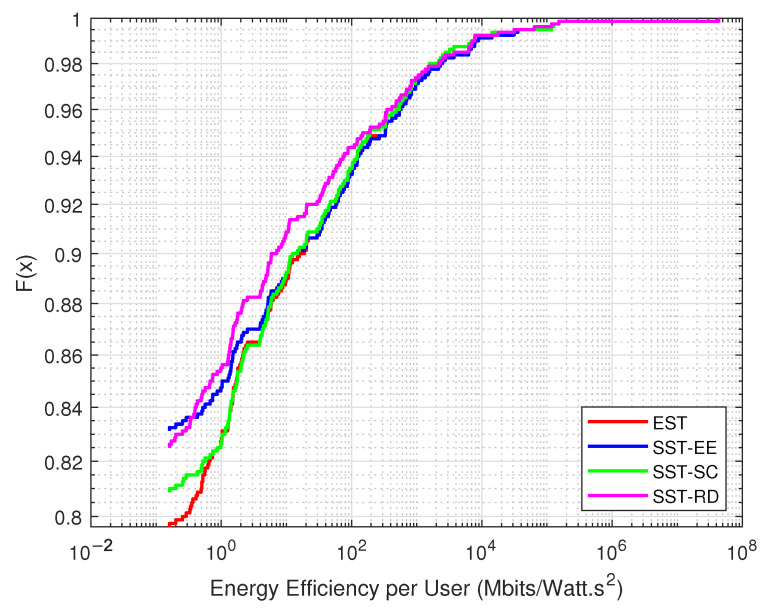
Variation in the distribution of the energy efficiency in the JSS.

**Figure 8 sensors-22-07744-f008:**
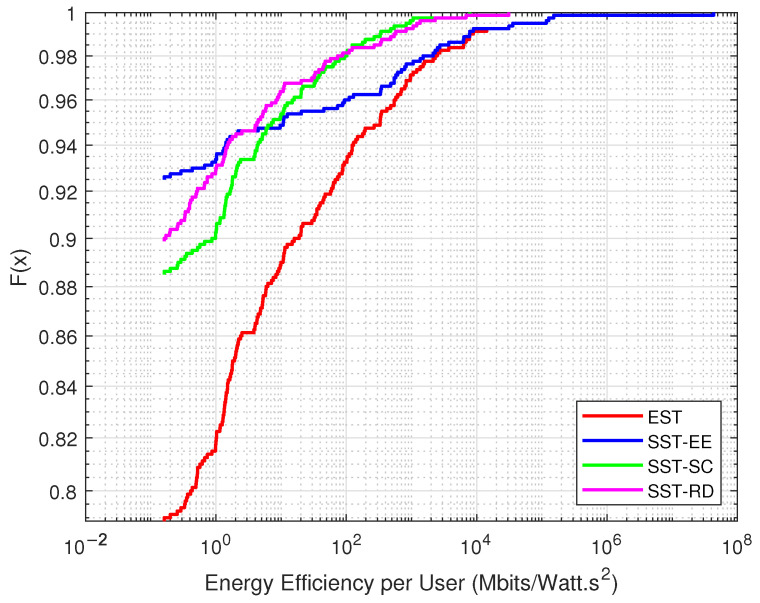
Variation in the distribution of the energy efficiency in the SSS.

**Figure 9 sensors-22-07744-f009:**
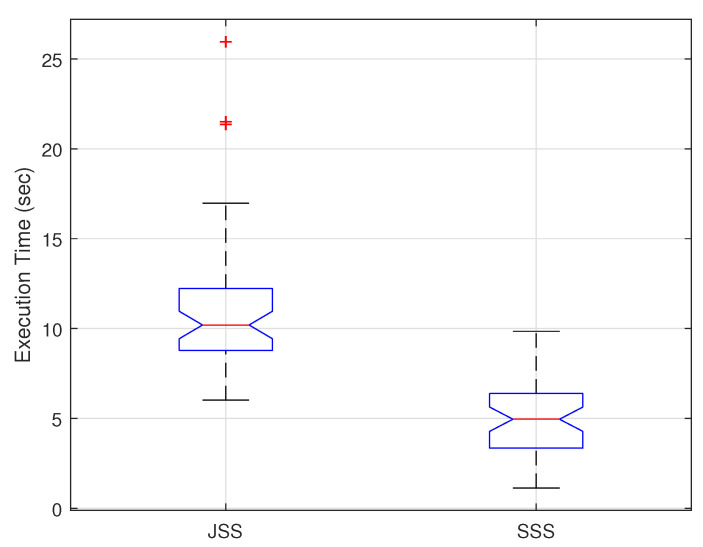
Variation in the execution time of the SSS and the JSS considering the SST-SC.

**Figure 10 sensors-22-07744-f010:**
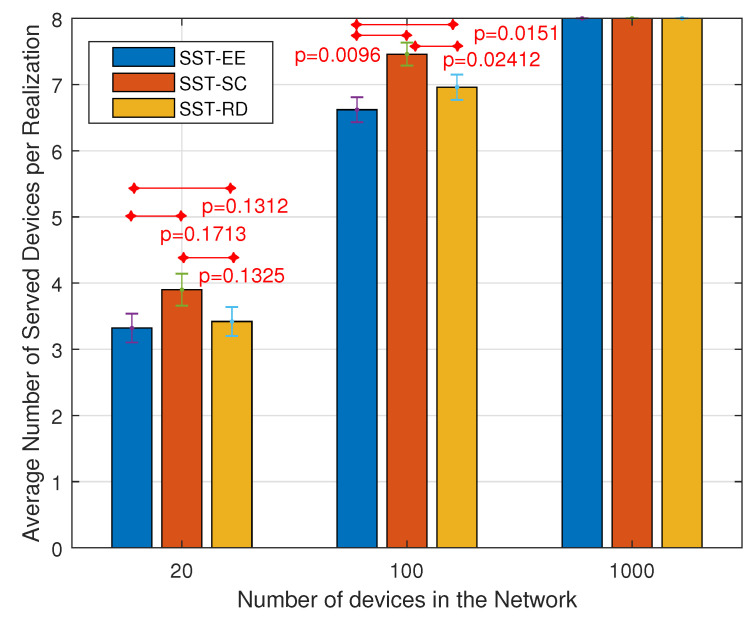
Variation in the average number of served devices per realization considering the JSS.

**Figure 11 sensors-22-07744-f011:**
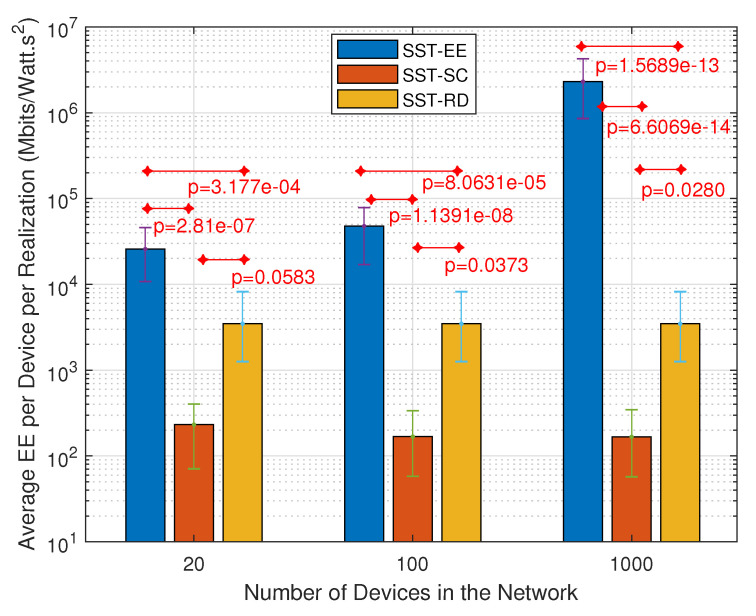
Variation in the average energy efficiency per device per realization considering the JSS.

**Figure 12 sensors-22-07744-f012:**
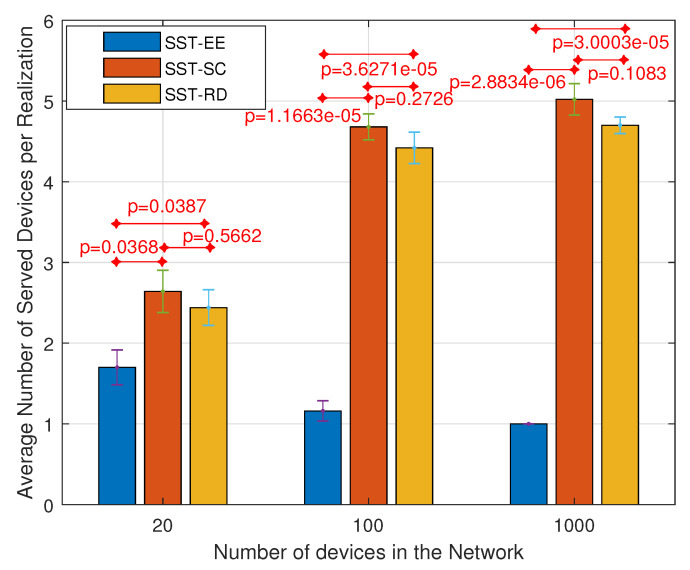
Variation in the average number of served devices per realization considering the SSS.

**Figure 13 sensors-22-07744-f013:**
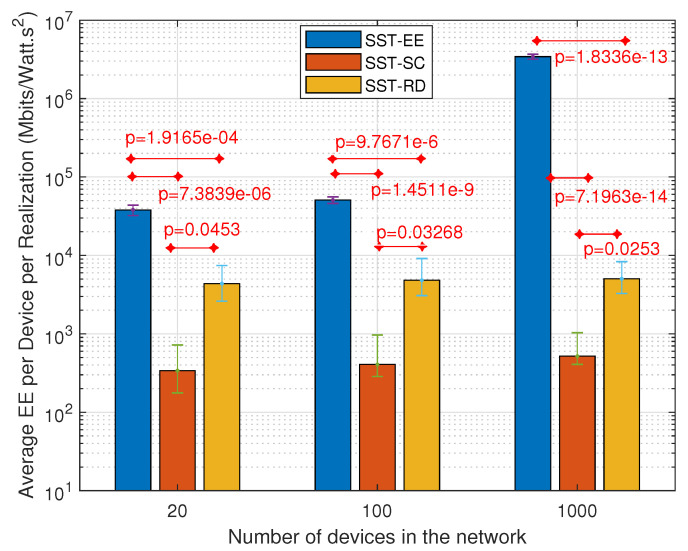
Variation in the average energy efficiency per device per realization considering the SSS.

**Table 3 sensors-22-07744-t003:** The Main Scheduling Fields.

Scheduling Field	Value
Subcarrier Indication Field ($I^{sc}$)	0∼63
Repetition Number Field ($I^{rep}$)	0∼7
Resource Assignment Field ($I^{RU}$)	0∼7
Modulation and Coding Scheme Field ($I^{MCS}$)	0∼13
Scheduling Delay Field ($I^{delay}$)	0∼3

**Table 4 sensors-22-07744-t004:** Characteristics of the different RU types for subcarrier spacing = 15 kHz.

Number of Tones (RU Type)	Number of Timeslots	RU Duration Time (ms)	RU Bandwidth (kHz)	Transmission Format
1-tone	16	8	15	Single-Tone
3-tone	8	4	45	Multi-Tone
6-tone	4	2	90	Multi-Tone
12-tone	2	1	180	Multi-Tone

**Table 5 sensors-22-07744-t005:** Subcarrier indication field values and its corresponding RU type and subcarrier sets.

Subcarrier Indication Value ($I^{sc}$)	RU Type	Subcarrier Set
0∼11	1-tone	$I^{sc}$
12∼15	3-tone	$3\cdot \left(I^{sc}-12\right)+\left\{0,1,2\right\}$
16∼17	6-tone	$6\cdot \left(I^{sc}-16\right)+\left\{0,1,2,3,4,5\right\}$
18	12-tone	{0,1,2,3,…,11}
19∼63		Reserved

**Table 6 sensors-22-07744-t006:** The supported uplink number of repetitions in NB-IoT.

Repetition Number Field ($I^{rep}$)	Repetition Number Value (*r*)
0	1
1	2
2	4
3	8
4	16
5	32
6	64
7	128

**Table 7 sensors-22-07744-t007:** The supported uplink resource assignment in NB-IoT.

Resource Assignment Field ($I^{RU}$)	Resource Assignment Value (*u*)
0	1
1	2
2	3
3	4
4	5
5	6
6	8
7	10

**Table 8 sensors-22-07744-t008:** The scheduling delay values supported for the uplink scheduling.

Scheduling Delay Field ($I^{delay}$)	Scheduling Delay Value ($k_{0}$) (ms)
0	8
1	16
2	32
3	64

**Table 9 sensors-22-07744-t009:** Abbreviations.

Symbol	Description
*N*	Number of devices
NP	NPDCCH Period
$R_{max}$	System parameter determining the NP length
*G*	System parameter determining the NP length
*L*	NP length
$r_{DCI}$	Number of times a DCI is repeated
ε	Number of candidates within an NP
$I^{sc}$	Subcarrier indication field
$I^{rep}$	Repetition number field
$I^{RU}$	Resource assignment field
$I^{MCS}$	Modulation and coding scheme field
$I^{delay}$	Scheduling delay field
*y*	RU types
*q*	Integer variable identifying the RU type
*i*	Integer variable identifying a device
*p*	Integer variable identifying an NP
$v_{p,i,q}$	Binary variable indicating if RU type *q* is allocated to device *i* in NP *p*
β	Number of subcarriers
*t*	Duration of the RU
$n^{sc}$	Subcarrier set
*r*	Repetition number value
*u*	Resource assignment value
*m*	Modulation and coding scheme level
$k_{0}$	Scheduling delay value
sb	first NPUSCH subframe
*n*	last NPDCCH subframe
*R*	Radius of the circular geographical area
$D_{i}$	Size of the uplink data of device *i* (in bits)
$p_{i}$	Transmission power of device *i*
$p^{min}$	Minimum Transmission power
$p^{max}$	Maximum Transmission power
$d_{i}$	Delay deadline of device *i* (in milliseconds)
$S_{rx}$	Received signal to noise and interference ratio
$S_{req}$	Minimum signal to noise ratio for successful decoding of the uplink transmission
*C*	CRC
$G_{r}$	Receiver Antenna Gain
$G_{t}$	Transmitter Antenna Gain
$L_{path}$	Path Loss
$N_{0}$	Noise
*w*	Bandwidth of the RU
$L_{shadow}$	Shadow Fading Effect
$\gamma \cdot L_{penetration}$	Penetration Loss
γ=1	Indoor Device
γ=0	Outdoor Device
*l*	Interference Level
$R_{b}$	Transmission bit rate (bits/s)
$B_{eff}$	Bandwidth efficiency of NB-IoT
$S_{eff}$	SNR efficiency of NB-IoT
σ	Estimation Error
$c_{1}$ and $c_{2}$	Constants obtained through link-level simulations
*B*	Transport block size (bits)
$B_{max}$	Maximum TBS
EE	Energy Efficiency
DR	Data Rate
CE	Consumed Energy
*s*	subframe index
*c*	subcarrier index
$x_{p,s,c,i}$	Binary allocation variable indicating if subcarrier *c* at subframe *s* in NP *p* is allocated to device *i*

**Table 10 sensors-22-07744-t010:** Definitions.

Introduced Scheduling Problems	
Successive Scheduling Problem (SSP)	Joint Scheduling Problem (JSP)
**Proposed Scheduling Schemes**	
Successive Scheduling Scheme (SSS)	Joint Scheduling Scheme (JSS)
**Device Selection Techniques**	
Exhaustive Search Technique (EST)	Sorting Score Technique (SST)

**Table 12 sensors-22-07744-t012:** The Simulation Parameters [1].

Parameter	Value
Maximum Transmit Power ($p^{max}$)	23 dBm
Minimum Transmit Power ($p^{min}$)	−40 dBm
Antenna Gain of Receiver ($G_{r}$)	9 dBi
Antenna Gain of Transmitter ($G_{t}$)	0 dBi
Thermal Noise Density ($N_{0}$)	−174 dBm/Hz
Cell Radius	4 km
Request Data Size (*D*)	50 bytes
Delay Value (*d*)	100 to 300 ms
Interference Margin	3 dBm
Number of NPDCCH Subframes ($R_{max}$)	256
Length of NPDCCH Period (*L*)	4096
Number of candidates in NP (ε)	8

## Data Availability

The data used to support the reported results of this paper are available upon request.

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
