# Peer review of "Energy-Efficient Uplink Scheduling in Narrowband IoT"

_sensors, 2022, doi:10.3390/s22207744_

Round 1
Reviewer 1 Report
Please see the attached file.

Author Response
Response to Reviewer 1 Comments
Dear Reviewer 1,
We would like to thank you for your time taken to review our manuscript and your constructive comments. We aim from this document to present a point-by-point response to your comments.
Point 1: It is suggested to use past tence to introduce the existing works.
Response 1: We thank you for this comment. We updated the “State of the art” section of the manuscript, where we presented each cited work using the past tense.
Point 2: There are many abbreviations in this paper, it is suggested to add a Table to make them more clear.
Response 2: We thank you for this suggestion. We added a table of notations, “Table 9. Notations”, at the end of section 5, clarifying all the symbols and abbreviations in the manuscript.
Point 3: The units of values in Table 1 should be given.
Response 3: Table 1, “The Main Scheduling Fields,” lists the scheduling fields carried by the DCI with their corresponding values. These values are unitless, as they are considered integer indicator variables.
Best regards,
Farah Yassine, on behalf of the authors
Reviewer 2 Report
This paper presented a detailed study of uplink scheduling in narrowband internet of things (NB-IoT) networks. As NB-IoT devices are in need for long battery lifetime, the authors aim to maximize the energy efficiency while satisfying the main requirements for NB-IoT devices. This paper deals with a hot area of investigation at the moment. I have looked at the mathematics and it looks sound. However, herein some minor issues need to address:
*Recommend dividing Section 1(Introduction) into 3 subsections (1.1. Background, 1.2. Motivation, and 1.3. Contributions) which improve the readability of the article.
*Fore symbols, recommend providing a symbols Table in the Appendix.
*Check Typos and grammar issues
Author Response
Response to Reviewer 2 Comments
Dear Reviewer 2,
We would like to thank you for your time taken to review our manuscript and your constructive comments. We aim from this document to present a point-by-point response to your comments.
Point 1: Recommend dividing Section 1 (Introduction) into 3 subsections (1.1. Background, 1.2. Motivation, and 1.3. Contributions) which improve the readability of the article.
Response 1: We thank you for pointing this out. We updated the “Introduction” section in the manuscript. We highlighted the motivation and contributions in the last three paragraphs in the “Introduction” section (marked in blue in the uploaded manuscript).
Point 2: Fore symbols, recommend providing a symbols Table in the Appendix.
Response 2: We added a table of notations, “Table 9. Notations,” at the end of section 5, clarifying all the symbols and abbreviations in the manuscript.
Point 3: Check Typos and grammar issues.
Response 3: Thank you for this comment. Please note that we checked the manuscript's grammar.
Best regards,
Farah Yassine, on behalf of the authors
Reviewer 3 Report
-The authors studied the NB-IoT uplink scheduling that aims to maximize the system’s energy efficiency while satisfying reliable, delay, and resource allocation constraints. They introduced two different device selection techniques to select the optimal devices to be served such that the system energy efficiency.
-The following major corrections are required:
Section 1: Introduction
-The introduction section is too general, and it introduces concepts that are well known about the uplink scheduling in narrowband internet of things (NB-IoT) networks. The introduction does not stimulate to go ahead with the remaining of the paper because it does not introduce any really new topic/solution. Furthermore, “the research motivation and challenges” into the introduction section is missing. Please rewrite this section.
Section 2: Related Works and Contributions
-In the related work section, the authors should describe of the research works about the uplink scheduling in NB-IoT networks, while some of the papers that should have be included are:
https://www.mdpi.com/1424-8220/22/8/2875
https://link.springer.com/chapter/10.1007/978-981-19-0390-8_96
https://onlinelibrary.wiley.com/doi/abs/10.1002/spe.2641
https://ieeexplore.ieee.org/abstract/document/9817417
https://onlinelibrary.wiley.com/doi/abs/10.1002/ett.3770
https://ieeexplore.ieee.org/abstract/document/9792254
--In addition, a conclusion of related work in the forms of a table in terms of evaluation tools, utilized techniques, performance metrics, datasets, advantages, and disadvantages could reconcile from other researchers work to the own one.
Section 3: General Overview on NB-IoT Scheduling
-What is the overhead (time complexity) proposed solution? Please provide a subsection to discuss about the overhead (time complexity) of the Algorithms 1-4.
-Please provided a real-world case study for better understanding the proposed approach in more details.
Section 7: Performance Evaluation
-The evaluation is incomplete. I would like to see an evaluation on the proposed solution in terms of execution time under different scenarios.
-The evaluation lacks the minimum rigor required for the scientific comparison of stochastic algorithms. Specifically, statistical tests of the hypothesis should be used to determine whether the differences shown in the figures are statistically significant or due to chance.
-Paper needs some revision in English. The overall paper should be carefully revised with focus on the language: especially grammar and punctuation.
-Overall, there are still some major parts that the authors did not explain clearly. Some additional evaluations are expected to be in the manuscript as well. As a result, I am going to suggest Major revision the paper in its present form.
Author Response
Please check the attached document for the author's reply.

Round 2
Reviewer 3 Report
-Unfortunately, the revised paper in its present suffers from several limitations and the flaws still exist.
Main points are:
Section 3: General Overview on NB-IoT Scheduling
-What is the overhead (time complexity) proposed solution? Please provide a subsection to discuss about the overhead (time complexity) of the Algorithms 1-4.
-Please provided a real-world case study for better understanding the proposed approach in more details.
Section 7: Performance Evaluation
-The evaluation is incomplete. The evaluation lacks the minimum rigor required for the scientific comparison of stochastic algorithms. Specifically, statistical tests of the hypothesis should be used to determine whether the differences shown in the figures are statistically significant or due to chance.
-Some key papers in the research area are left out. Some of the related papers that should have be included are:
https://www.mdpi.com/1424-8220/22/8/2875
https://link.springer.com/chapter/10.1007/978-981-19-0390-8_96
https://onlinelibrary.wiley.com/doi/abs/10.1002/spe.2641
https://ieeexplore.ieee.org/abstract/document/9817417
https://onlinelibrary.wiley.com/doi/abs/10.1002/ett.3770
https://ieeexplore.ieee.org/abstract/document/9792254
-Overall, there are still some major parts that the authors did not explain clearly. Some additional evaluations are expected to be in the manuscript as well. As a result, I am going to suggest Rejecting the paper in its present form.
Author Response
I would like to thank you for the provided comments. Please check the response to your comments in the attached document.

Round 3
Reviewer 3 Report
| Thanks to authors for the detailed response and additions I read through the comments and skimmed the revised PDF, The updates did improve the paper a lot. I would be happy to recommend this paper for publication |